ecology/taxonomy and systematics/molecular biology

hydrothermal vents, stable isotopes, taxonomy, crustaceans, life history, trophic shift

**Author for correspondence:**
Florence Pradillon
e-mail: florence.pradillon@ifremer.fr

# Integrative taxonomy revisits the ontogeny and trophic niches of *Rimicaris* vent shrimps

Pierre Methou[1,2], Loïc N. Michel[2], Michel Segonzac[3], Marie-Anne Cambon-Bonavita[1] and Florence Pradillon[2]

[1]Ifremer, Univ Brest, CNRS, Laboratoire de Microbiologie des Environnements Extrêmes, UMR 6197, F-29280 Plouzané, France
[2]Ifremer, Centre Brest, Laboratoire Environnement Profond (REM/EEP/LEP), ZI de la pointe du Diable, F-29280 Plouzané, France
[3]Institut de Systématique, Évolution, Biodiversité (ISYEB), Muséum national d'Histoire naturelle, CNRS, Sorbonne Université, EPHE, case postale 53, 57 rue Cuvier, F-75231 Paris cedex 05, France

 PM, 0000-0001-5454-1775; LNM, 0000-0003-0988-7050; M-AC-B, 0000-0002-4076-0472; FP, 0000-0002-6473-6290

Among hydrothermal vent species, *Rimicaris exoculata* is one of the most emblematic, hosting abundant and diverse ectosymbioses that provide most of its nutrition. *Rimicaris exoculata* co-occurs in dense aggregates with the much less abundant *Rimicaris chacei* in many Mid-Atlantic Ridge vent fields. This second shrimp also houses ectosymbiotic microorganisms but has a mixotrophic diet. Recent observations have suggested potential misidentifications between these species at their juvenile stages, which could have led to misinterpretations of their early-life ecology. Here, we confirm erroneous identification of the earliest stages and propose a new set of morphological characters unambiguously identifying juveniles of each species. On the basis of this reassessment, combined use of C, N and S stable isotope ratios reveals distinct ontogenic trophic niche shifts in both species, from photosynthesis-based nutrition before settlement, towards a chemosynthetic diet afterwards. Furthermore, isotopic compositions in the earliest juvenile stages suggest differences in larval histories. Each species thus exhibits specific early-life strategies that would, without our re-examination, have been interpreted as ontogenetic variations. Overall, our results provide a good illustration of the identification issues persisting in deep-sea ecosystems and the importance of integrative taxonomy in providing an accurate view of fundamental aspects of the biology and ecology of species inhabiting these environments.

# 1. Introduction

In recent years, several biodiversity hotspots have been discovered in the deep ocean, with ecosystems such as hydrothermal vents, cold seeps, canyons or seamounts [1]. Without endogenous photosynthesis, their high productivity is supported by other mechanisms, like chemosynthesis, or through physical processes enhancing transport and deposition of particles sinking from the surface. In hydrothermal vents, it is mainly the chemical disequilibria generated by the mixing of cold oxygenated seawater with hot reduced fluid emissions that sustains a wide range of chemosynthetic microorganisms, which in turn support food webs [2].

Most of these ecosystems remain poorly explored and could contain one of the largest reservoirs of undescribed species on our planet. Indeed, sampling in these remote places is often insufficient to capture the full range of spatial and temporal biological variability, even for the most well-known species [3]. Although often disregarded, developmental plasticity in response to a variable environmental context can have a major influence on the morphology of some organisms. This has led to many taxonomic controversies, since morphology alone can be misleading. Ontogenic morphological variability is also a great source of taxonomic bias. Small orange/reddish alvinocaridid shrimps found in hydrothermal vents close to dense swarms of *Rimicaris exoculata* [4] are a classic example of such variation between life stages. Simultaneously described as *Iorania concordia* and *R. aurantiaca* by two concomitant studies [5,6], confusion about the taxonomic status of these specimens was dispelled by molecular studies [7]. These shrimps were, in fact, revealed to be juvenile stages of *R. exoculata*, which was further supported by the absence of sexually mature individuals among these small orange morphotypes [7]. These juveniles were later described by Komai and Segonzac [8] as four distinct stages corresponding to the different ontogenic transitions of *R. exoculata*, the fourth ones being also considered as the subadult stage, morphologically similar to adult individuals but not yet sexually mature.

Beyond the taxonomy, this re-description of *R. exoculata* life stages had a drastic impact on the understanding of the life-history and trophic ecology of this species. In hydrothermal vents, food web structure and trophic interactions are often inferred from stable isotope markers. Different stable isotope ratios can provide complementary ecological information. For example, the large difference in sulfur isotopic ratios ($\delta^{34}$S) between seawater sulfate and vent fluid sulfides has been used to discriminate between organic matter of photosynthetic (approx. 16‰ to 21‰) and chemosynthetic origin (−9‰ to 10‰) [9,10]. Moreover, differences in carbon isotopic ratios ($\delta^{13}$C) are attributed to carbon sources fixed through the reductive tricarboxylic acid (rTCA) cycle (2‰ to −14‰) or the Calvin–Benson–Bassham (CBB) cycle (−22‰ to −30‰) [11]. Additionally, nitrogen isotopic ratios ($\delta^{15}$N) are generally used to infer the trophic position of a species within the food web [12].

*Rimicaris exoculata* adults depend for their nutrition on the dense and diversified chemosynthetic episymbiotic communities hosted in their enlarged cephalothorax cavities [13–15]. Despite unusually high $\delta^{15}$N values for an animal feeding directly on chemosynthetic bacteria, $\delta^{13}$C composition of *R. exoculata* adults [16,17] is coherent with a nutrition mainly based on primary producers using the rTCA carbon fixation pathway. Moreover, direct nutritional transfer of light organic carbon compounds occurs between the host and its symbionts [18]. In the small orange shrimps, however, lipids and isotopic compositions have suggested a diet of photosynthetic origin [17,19]. Thus, without the taxonomic reassignment of orange shrimps as *R. exoculata* juveniles, the ontogenic trophic shift from photosynthesis-based nutrition outside the vent field to a feeding mode dependent on chemosynthetic symbionts would have been interpreted as distinct feeding habits of two different species. Furthermore, regarding the life history, isotopic data argue for a long planktotrophic larval dispersal in *R. exoculata* before settlement as juveniles on a hydrothermal vent field and transition to a chemosynthetic feeding mode [20].

Recently, small alvinocaridid juveniles living in dense patchy aggregations, defined as nursery habitats, were collected close to diffuse low-temperature emissions arising from bottom cracks at the TAG vent field [21]. These juveniles contained red/orange storage lipids and looked like the first juvenile stage of *R. exoculata* (also called stage A juvenile) in terms of size and overall morphology [8]. Surprisingly, molecular identification using the COI gene revealed that they were all affiliated to *R. chacei* [21]. *Rimicaris chacei* co-occurs with *R. exoculata* but is presumed to be less abundant, as adults are much less conspicuous on vents [15]. Unlike *R. exoculata*, *R. chacei* has a mixotrophic diet, relying on both symbiotrophy, bacterivory and scavenging [15,17,22]. This feeding plasticity is supported by both their anatomical characteristics and their symbiotic communities, which are closely related to

those found in *R. exoculata*, but less developed [15,22]. This mixotrophy has been hypothesized to be the result of competition for space with *R. exoculata* that would maintain *R. chacei* at a distance from vent fluids essential for fuelling their symbionts [22]. Given the potential misidentification of *Rimicaris* juvenile stages mentioned earlier, a re-examination of juvenile stages in terms of species assignment and isotopic compositions should be warranted to reassess potential ontogenic shifts in post-recruitment stages. In addition, comparison between the two *Rimicaris* species should provide a better understanding of how symbiosis and feeding habits shift from juvenile recruitment to adult stages, and of the mechanisms underlying the coexistence of the two species.

Here, we reassessed the taxonomic status of the juveniles collected from nursery habitats and from nearby adult aggregates using coupled morphological analyses and DNA barcoding. Then, using stable isotopes of multiple elements, we explored the ecological consequences of this reassessment on our understanding of the trophic behaviour of these two species and, ultimately, the implications it has for their life histories.

# 2. Material and methods

## 2.1. Sample collection

Alvinocaridid shrimps were collected from the Snake Pit (23°22′ N; 44°57′ W, 3460 m depth) and TAG (26°08′ N; 44°49.5′ W, 3630 m depth) vent fields on the Mid-Atlantic Ridge (MAR) using a suction sampler manipulated by the human operated vehicle (HOV) *Nautile*. Carapace length (CL) of each individual was measured with vernier callipers from the posterior margin of the ocular shield (or eye socket) to the mid-point of the posterior margin of the carapace, with an estimated precision of 0.1 mm. About 205 small alvinocaridid juveniles collected mostly from nurseries habitats (177 individuals) but also from dense *R. exoculata* aggregates (28 individuals) during the HERMINE cruise (16 March–27 April 2017) were stored in 80% ethanol for later detailed morphological analysis and individual barcoding. *Rimicaris* shrimps of both species and all life stages were collected during the BICOSE 2 cruise (26 January–10 March 2018). This second collection was used to assess the onset of sexual differentiation (OSD) in *R. chacei*, based on formalin fixed individuals (130 specimens) and to perform isotopic analyses using −80°C frozen tissue (208 specimens, detailed per species and per life stages in figures 3 and 4).

## 2.2. Morphological analysis

Juvenile specimens were examined in detail under a stereomicroscope (Leica MS80) to assess their individual stage and/or species according to morphological descriptions of stage A juveniles of *R. exoculata* and juveniles of *R. chacei* [8]. They were expected to be distinct from each other according to five morphological characters (detailed in electronic supplementary material, figure S1) selected from a published taxonomic description [8].

## 2.3. Genetic identification

About 64 juveniles selected from the 2017 collection, and representing all morphological types identified, were individually genetically characterized. DNA was extracted from pieces of telson or muscle with the DNeasy Blood & Tissue kit (Qiagen) following manufacturer's instructions. To avoid amplification of potential mitochondrial pseudogenes, new primers Cari-COI-1F (5′-GCAGTCTRGYGTCTTAATTT CCAC-3′) and Cari-COI-1R (5′-GCTTCTTTTTTACCRGATTCTTGTC-3′) were designed, based on an alignment of the mitochondrial genome of six alvinocaridid species [21]. These primers amplified a 769 bp fragment in the 5′ region of the COI gene, including the section amplified by the universal primers LCOI1490 and HCOI2198 [23]. Amplicons were sequenced (Sanger) on both strands (Macrogen). To confirm taxonomic assignment, each individual sequence was blasted against the NCBI database. All sequences have been deposited in GenBank under accession numbers MT270699 to MT270782. The haplotype network was obtained and edited with PopART [24] using a median-joining approach [25], with *R. exoculata* (KP759507, HM125956), *R. chacei* (KC840930, KT210445) and *Mirocaris fortunata* (KP759434, KT210455) as reference sequences. Although larger juveniles assigned to later *R. exoculata* juveniles (stage B, C [8]) were not the focus of this study, 20 of them were also barcoded to confirm taxonomic assignations of all juvenile stages encountered in our samples.

## 2.4. Stable isotope analysis

Abdominal muscle samples were oven-dried to constant mass at 50°C (>48 h) before being ground into a homogeneous powder using a mortar and pestle. Stable isotope ratio measurements were performed by continuous flow–elemental analysis–isotope ratio mass spectrometry (CF-EA-IRMS) at University of Liège (Belgium), using a vario MICRO cube C-N-S elemental analyser (Elementar Analysensysteme GMBH, Hanau, Germany) coupled to an IsoPrime100 isotope ratio mass spectrometer (Isoprime, Cheadle, United Kingdom). Isotopic ratios were expressed in ‰ using the widespread $\delta$ notation [26] and relative to the international references Vienna Pee Dee Belemnite (for carbon), Atmospheric Air (for nitrogen) and Vienna Canyon Diablo Troilite (for sulfur). Sucrose (IAEA-C-6; $\delta^{13}C = -10.8 \pm 0.5‰$; mean ± SD), ammonium sulfate (IAEA-N-1; $\delta^{15}N = 0.4 \pm 0.2‰$; mean ± s.d.) and silver sulfide (IAEA-S-1; $\delta^{34}S = -0.3‰$) were used as primary analytical standards. Sulfanilic acid (Sigma-Aldrich; $\delta^{13}C = -25.6 \pm 0.4‰$; $\delta^{15}N = -0.13 \pm 0.4‰$; $\delta^{34}S = 5.9 \pm 0.5‰$; means ± s.d.) was used as a secondary analytical standard. Standard deviations on multi-batch replicate measurements of secondary and internal laboratory standards (seabass muscle), analysed interspersed with samples (one replicate of each standard every 15 analyses), were 0.2‰ for $\delta^{13}C$, 0.3‰ for $\delta^{15}N$ and 0.4‰ for $\delta^{34}S$.

## 2.5. Statistical analysis and data processing

All analyses were performed in the R v. 3.5.1 statistical environment (R Core Team 2018). Alvinocaridid shrimps were clustered according to species, life stage and vent field origin. Visual examination of our dataset and Shapiro–Wilk normality tests revealed that none of the three isotopic ratios ($\delta^{13}C$, $\delta^{15}N$ and $\delta^{34}S$) followed a Gaussian distribution. Therefore, non-parametric tests were used for intergroup comparisons (Wilcoxon test when two groups were compared, Kruskal–Wallis test followed by post hoc pairwise Wilcoxon tests when three or four groups were compared; detailed $p$-values for all these tests are given in electronic supplementary material, table S1). Ecological niches were explored using the SIBER (electronic supplementary material, table S2) (Stable Isotope Bayesian Ellipses in R; see details in supplementary material) v. 2.1.4 package [27]. Two separate sets of standard ellipses were constructed: one using the $\delta^{13}C$ and $\delta^{15}N$ data and another using the $\delta^{13}C$ and $\delta^{34}S$ data.

# 3. Results

## 3.1. Species identification

Carapace length (CL) of the 205 small juveniles examined ranged between 3.8 and 8.7 mm. Using morphological features defined by Komai and Segonzac ([8], summarized in electronic supplementary material, figure S1), 128 juveniles were either identified as *R. chacei* juveniles or as *R. exoculata* stage A juveniles, while the 57 remaining exhibited chimeric morphologies.

Individual barcoding of 64 individuals randomly selected from all observed morphologies showed that they were all *R. chacei*, except one specimen with a conspicuous antennal spine that belonged to *Mirocaris fortunata* (figure 1).

Juveniles genetically identified as *R. chacei* were separated in two different stages. The first, further on called *R. chacei* stage A juvenile (figure 2a) (and gathering all individuals initially classified as *R. exoculata* stage A juveniles), was characterized by a produced triangular rostrum and a straight pterygostomial angle (figure 2b and table 1). The second stage looked much more like an adult of *R. chacei*, with a round rostrum, produced pterygostomial angle, as well as much less conspicuous orange lipid storage dorsally, oval shaped eyes and slightly larger sizes (figure 2c and table 1). Examination of gonad development in 130 *R. chacei* with CL ranging from 4 to 12.5 mm indicated that the onset of sexual differentiation (OSD) occurs at CL 5.98 mm (electronic supplementary material, figure S2), suggesting that our second stage was indeed a mixture of small adults and subadults. We thus define a stage B juvenile—or subadult—for *R. chacei* juveniles with adult-like morphology and CL sizes below OSD (table 1).

*Rimicaris exoculata* juveniles were then left with three stages only instead of four, named stage B, C and D (or subadult) by Komai & Segonzac [8], and here further renamed stage A, B and C (or subadult), respectively. With markedly larger sizes, these juveniles were easily identified following descriptions by Komai & Segonzac, which was further confirmed by barcoding (figure 1).

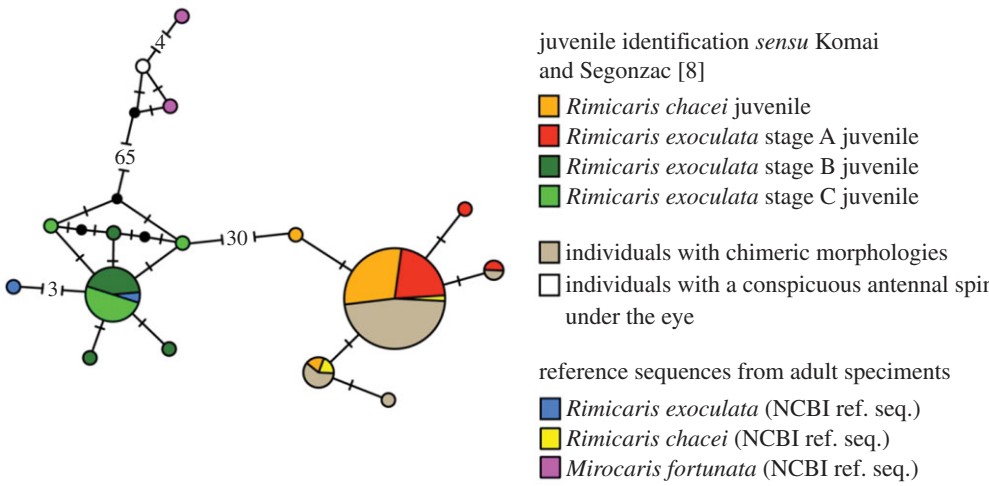

juvenile identification *sensu* Komai and Segonzac [8]

■ *Rimicaris chacei* juvenile (orange)
■ *Rimicaris exoculata* stage A juvenile (red)
■ *Rimicaris exoculata* stage B juvenile (dark green)
■ *Rimicaris exoculata* stage C juvenile (green)

■ individuals with chimeric morphologies
□ individuals with a conspicuous antennal spine under the eye

reference sequences from adult speciments

■ *Rimicaris exoculata* (NCBI ref. seq.)
■ *Rimicaris chacei* (NCBI ref. seq.)
■ *Mirocaris fortunata* (NCBI ref. seq.)

**Figure 1.** COI haplotype network obtained with a median-joining method in PopART from sequences of the different small alvinocaridid juveniles (*n* = 64) that were analysed morphologically, as well as of some larger juveniles (*R. exoculata* stages B and C *sensus* Komai & Segonzac [8]) (*n* = 20), with additional COI reference sequences of alvinocaridid adults from the MAR (*n* = 6). Sizes of coloured circles indicate relative haplotype frequencies. Number in brackets indicates the mutational steps between haplotypes.

## 3.2. Stable isotope ratios of carbon, nitrogen and sulfur through *Rimicaris* shrimp life stages

Life stages of both *R. exoculata* and *R. chacei* showed significantly distinct $\delta^{13}$C values (Kruskal–Wallis, $p < 0.05$), with a general trend of $^{13}$C-enrichment in later stages for both TAG and Snake Pit populations (figure 3a). Moreover, $^{13}$C-enrichment was progressive, with significant differences between almost all successive life stages of each species in each vent field (pairwise Wilcoxon test, $p < 0.05$) (figure 3a). Only $\delta^{13}$C differences between stage A and stage B juveniles and between stage B juveniles and subadults of *R. exoculata* from TAG, as well as between subadults and adults of *R. chacei* from Snake Pit, were not significant (pairwise Wilcoxon test, $p > 0.05$) (figure 3a). Differences could also be observed when similar life stages were compared between the species. Stage A juveniles of *R. exoculata* showed significantly less negative $\delta^{13}$C values than *R. chacei* stage A juveniles both at TAG (*R. exoculata*: $\delta^{13}$C = −17.8 ± 0.6‰; *R. chacei*: $\delta^{13}$C = −19.0 ± 0.6‰; Wilcoxon test, $p < 0.05$) and Snake Pit (*R. exoculata*: $\delta^{13}$C = −17.3 ± 0.3‰; *R. chacei*: $\delta^{13}$C = −18.6 ± 0.8‰; Wilcoxon test, $p < 0.05$) (figure 3a). The same pattern of less negative $\delta^{13}$C values in *R. exoculata* persisted in adults and, to a lesser extent, in subadults, in both fields (figure 3a).

Life stages of both species showed significantly different $\delta^{15}$N values (Kruskal–Wallis, $p < 0.05$), with a general trend of $^{15}$N-enrichment in later stages, for both vent fields (figure 3b). However, significant differences were only observed between adults and subadults of both species (pairwise Wilcoxon test, $p < 0.05$), and between stage A juveniles and subadults of *R. exoculata* at Snake Pit (pairwise Wilcoxon test, $p < 0.05$). Interspecies comparison at similar life stages did not reveal any significant differences (Wilcoxon test, $p > 0.05$) except between adult stages, where there were slightly lower $\delta^{15}$N values for *R. exoculata* at TAG (*R. exoculata*: $\delta^{15}$N = 7.1 ± 0.2‰; *R. chacei*: $\delta^{15}$N = 8.1 ± 1.1‰; Wilcoxon test, $p < 0.05$), but slightly higher ones at Snake Pit (*R. exoculata*: $\delta^{15}$N = 7.4 ± 0.4‰; *R. chacei*: $\delta^{15}$N = 6.5 ± 1.2‰; Wilcoxon test, $p < 0.05$).

Significant differences in $\delta^{34}$S were also found among life stages of *R. exoculata* and *R. chacei* in each vent field (Kruskal–Wallis, $p < 0.05$), with marked $^{34}$S-depletion towards later stages. $\delta^{34}$S values decreased progressively, with significant differences between successive life stages of *R. exoculata* and *R. chacei* in each vent field (pairwise Wilcoxon test, $p < 0.05$) (figure 3c). Interestingly, interspecific comparisons between similar life stages did not show any significant differences in $\delta^{34}$S (Wilcoxon test, $p > 0.05$), except between *R. exoculata* and *R. chacei* adults from TAG (*R. exoculata*: $\delta^{34}$S = 8.5 ± 0.7‰; *R. chacei*: $\delta^{34}$S = 11.3 ± 1.2‰; Wilcoxon test, $p < 0.05$) (figure 3c).

## 3.3. Stable isotope ellipses: variation between *Rimicaris* life stages

SIBER analysis revealed that the core isotopic niches of *R. exoculata* juveniles were markedly separated from those of adults for both the carbon versus sulfur ellipses (figure 4a and b) and the carbon versus

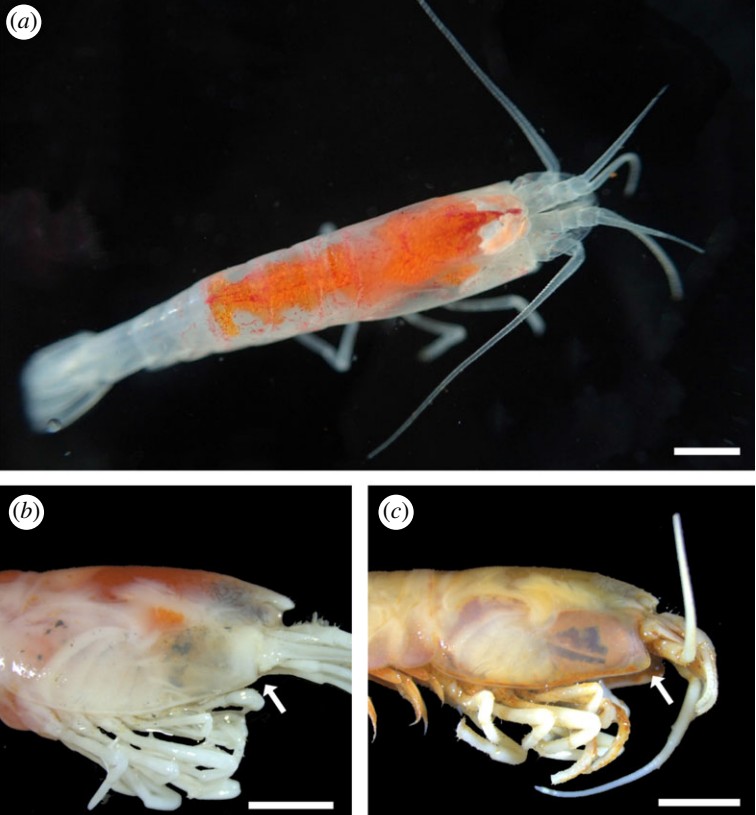

**Figure 2.** R. chacei juvenile identification according to the revised classification. (a) Freshly collected small alvinocaridid juvenile previously identified as a R. exoculata stage A juvenile but reassigned as a R. chacei stage A juvenile in our present work. Scale bar = 2 mm. (b) R. chacei stage A juvenile characterized by its straight pterygostomial angle (white arrow) and its pronounced rostrum. Scale bar = 2 mm. (c) R. chacei stage B juvenile (=subadult) characterized by its produced pterygostomial angle (white arrow) and its reduced rostrum. Scale bar = 2 mm.

nitrogen ellipses (figure 4c and d). Isotopic niche segregation could also be seen between juvenile stages with, in most cases, absolutely no overlap between successive stages from the same vent site for carbon versus sulfur ellipses (figure 4a and b). The only exception was at TAG where a very small overlap of $0.02‰^2$ (i.e. 2.6% of the smaller ellipse area, see electronic supplementary material, table S2 for details of ellipse overlaps) was found between subadults (turquoise, figure 4a) and stage B juveniles (light green, figure 4a). A similar pattern could be observed with the carbon versus nitrogen ellipses, for which overlap was more frequent but always low (figure 4c and d). At TAG, ellipses of stage A and stage B juveniles overlapped by less than $0.06‰^2$ (i.e. 7.9% of the smaller ellipse area), and those of stage B juveniles and subadults overlapped by $0.04‰^2$ (i.e. 5.8% of the smaller ellipse area; figure 4c). In the same way, at Snake Pit, ellipses of stage A and stage B juveniles overlapped by only $0.05‰^2$ (i.e. 11.6% of smaller ellipse area), and no overlap could be observed between ellipses of stage B juveniles and subadults (figure 4d).

For R. chacei, the same pattern of a clear isotopic niche segregation was observed between each life stage, except between subadults and adults at Snake Pit (figure 4). The overlap between these two life stages was quite low when carbon versus sulfur ellipses were examined ($0.24‰^2$, i.e. 0.3% of the smaller ellipse area; figure 4b), but more important with carbon versus nitrogen ellipses ($1.97‰^2$, i.e. 76.2% of the smaller ellipse area; figure 4d).

Interspecies comparisons of Rimicaris life stages revealed a strong isotopic niche segregation between species for carbon versus sulfur ellipses (figure 4a and b). At TAG, no interspecific overlap could be observed, except between R. exoculata and R. chacei stage A juveniles, for which it was present but almost null ($0.02‰^2$, i.e. 3.1% of the smaller ellipse area; figure 4a). Similarly, at Snake Pit, limited interspecific overlap was observed (figure 4b). It was only seen between R. chacei subadults and R. exoculata stage B, and between R. chacei and R. exoculata subadults. Carbon versus nitrogen ellipses overlapped more, although overlap was notably absent for both vent fields between stage A juveniles and between adults of the two species.

**Table 1.** Revised classification of alvinocaridid juveniles from Mid-Atlantic Ridge vents.

| this study | after Komai & Segonzac [8] | size range (CL in mm) | lipid storage | morphological characteristics |
|---|---|---|---|---|
| R. chacei stage A (juvenile) | R. exoculata stage A (juvenile) | 3.7–5.4 | yes | carapace not inflated and straight, pterygostomial angle straight, pronounced rostrum, rounded eyes |
| R. chacei stage B (subadult) | R. chacei (juvenile) | 4–5.98 | reduced | carapace not inflated and straight, pterygostomial angle produced, reduced rostrum, oval-shaped eyes |
| R. chacei (adult) | R. chacei (adult) | 5.98–21.7 | no | carapace not inflated and straight, pterygostomial angle produced, reduced rostrum, oval-shaped eyes |
| R. exoculata stage A (juvenile) | R. exoculata stage B (juvenile) | 5.7–9.7 | yes | carapace not inflated but tapered, oval-shaped eyes clearly separated, pronounced rostrum |
| R. exoculata stage B (juvenile) | R. exoculata stage C (juvenile) | 6.2–9.9 | yes | carapace more inflated, eyes closer to each other's, in fusion, rostrum extremely reduced |
| R. exoculata stage C (subadult) | R. exoculata stage D (subadult) | 7.6–9.9 | reduced | carapace strongly inflated, ocular plate formed, rostrum absent |
| R. exoculata (adult) | R. exoculata (adult) | 9.9–24.4 | no | carapace strongly inflated, ocular plate formed, rostrum absent |
| M. fortunata | M. fortunata | 3.5–10.7 | no | pterygostomial angle straight, small spine on the carapace under the eye, pronounced rostrum |

# 4. Discussion

## 4.1. Identification of *Rimicaris exoculata* and *Rimicaris chacei* juveniles

Using detailed morphological analyses and individual barcoding, we propose a revision of the life cycle of *Rimicaris* species, with *R. chacei* having two juvenile stages, while *R. exoculata* has three juvenile stages instead of four as described until now in the literature [8] (figure 5*a* and *b*). These newly defined stages can be distinguished morphologically according to the shape of the rostrum and of the pterygostomial angle, but also differ in their amount of storage lipids, and eye shape (table 1, figure 2*b* and *c*).

At MAR vent fields, with four alvinocaridid species potentially coexisting on the same sites, morphological identification of juveniles remains, however, challenging. In the *Rimicaris* genus, stage A *R. exoculata* juveniles have a rounded rostrum and produced pterygostomial angle, similar to the subadult (=juvenile stage B) of *R. chacei*. Additional features, such as a larger size (>7 mm CL), more abundant and conspicuous dorsal lipid reserves, and an anteriorly tapered carapace in *R. exoculata* stage A juveniles allow differentiation from *R. chacei* subadults (table 1; electronic supplementary material, figure S3). In addition, while the former is usually found near adult conspecifics, the latter occurs near diffusions or at the periphery of large shrimp aggregates [21,28]. Stage A *R. chacei* juveniles can also be confounded with small *Mirocaris fortunata* (figure 5*c*) that are often observed very close or mixed with R. *chacei* in several vent fields. In freshly collected specimens, red/orange lipid reserves proved a reliable criterion to discriminate between these two species. However, this coloration of lipids quickly fades after a few weeks of ethanol storage, thus becoming misleading. The front part of the carapace, which features a straight pterygostomial angle in both cases but bears a

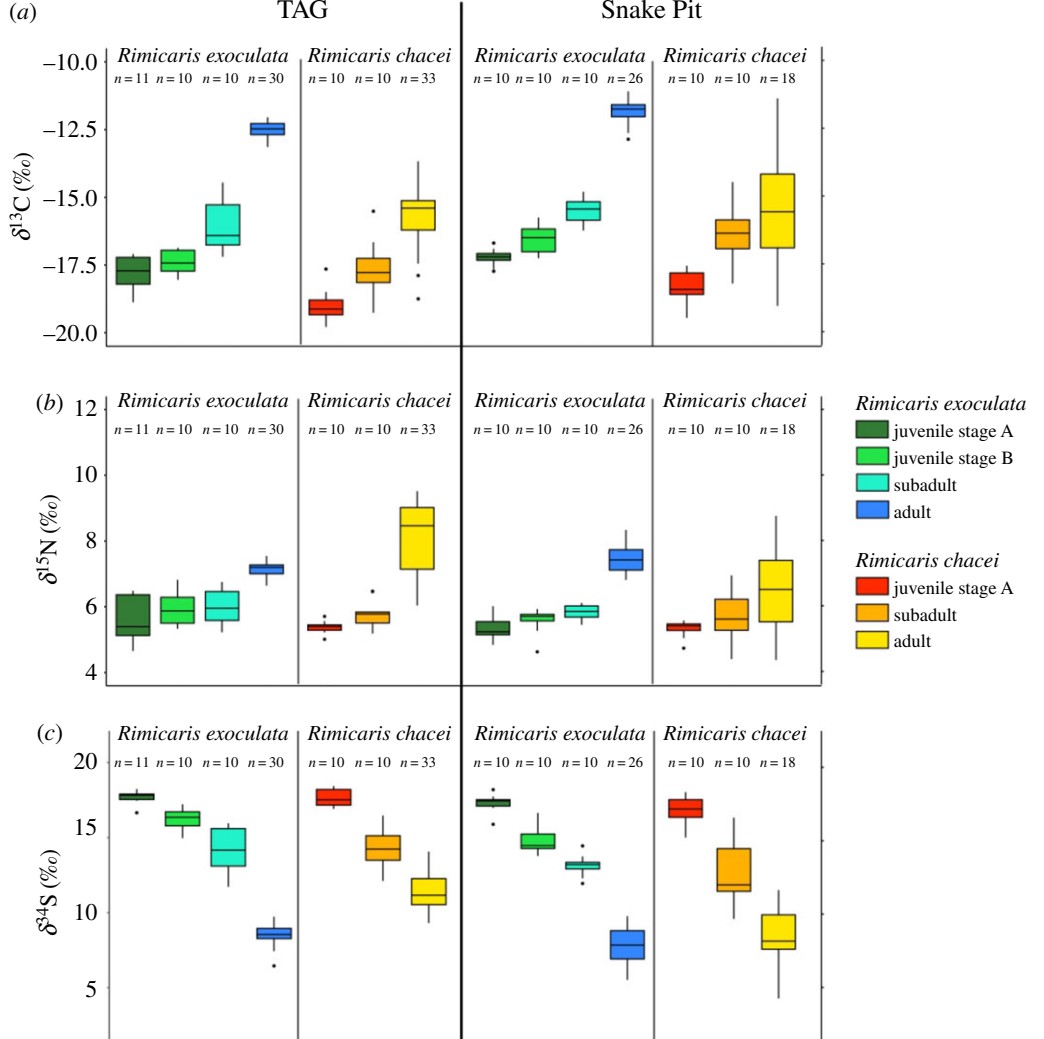

**Figure 3.** Isotopic ratios of *R. exoculata* and *R. chacei* from TAG and Snake Pit at different life stages for (*a*) carbon, (*b*) nitrogen and (*c*) sulfur.

prominent spine below the eye in *M. fortunata*, could be an additional criterion for morphological distinction. For correct identification in case of doubt, we strongly recommend the use of COI barcoding.

## 4.2. Isotopic niches suggest contrasting resource use in *Rimicaris exoculata* and *Rimicaris chacei*

Adults of each species show distinct isotopic niches, suggesting that their use of the resources offered by the vents differ. This absence of overlap is largely related to a significant $^{13}$C-enrichment in *R. exoculata*. According to several studies [13,14,22], symbionts using both the rTCA (affiliated to the *Campylobacterota*, formerly named *Epsilonproteobacteria* [29]) and the CBB (affiliated to the *Gammaproteobacteria* or *Zetaproteobacteria*) carbon-fixing pathways are hosted in the cephalothorax of *R. exoculata* and *R. chacei*. The main nutrient supply of *R. exoculata* adults appears to be their dominant *Campylobacterota* symbionts, as their $\delta^{13}$C values are clearly in the range of rTCA-fixed carbon sources [11]. This suggests that other symbionts only have a minor contribution to their diet. By contrast, $\delta^{13}$C values of *R. chacei* adults suggest a mixed diet, based on multiple carbon sources. This could be related to a greater contribution from the *Gammaproteobacteria* or other CBB-fixing symbionts than from the *Campylobacterota* compared with *R. exoculata*. However, *R. chacei* adults could also depend on food sources other than their symbionts. Although $\delta^{34}$S values of *R chacei* adults from Snake Pit are purely in the range of a chemosynthetic diet, those of *R. chacei* from TAG correspond indeed to a diet of mixed origins (figure 4). Previous observations of organic matter in the stomach contents of *R. chacei* also support mixotrophic nutrition [15]. One of the most striking divergences observed at TAG compared

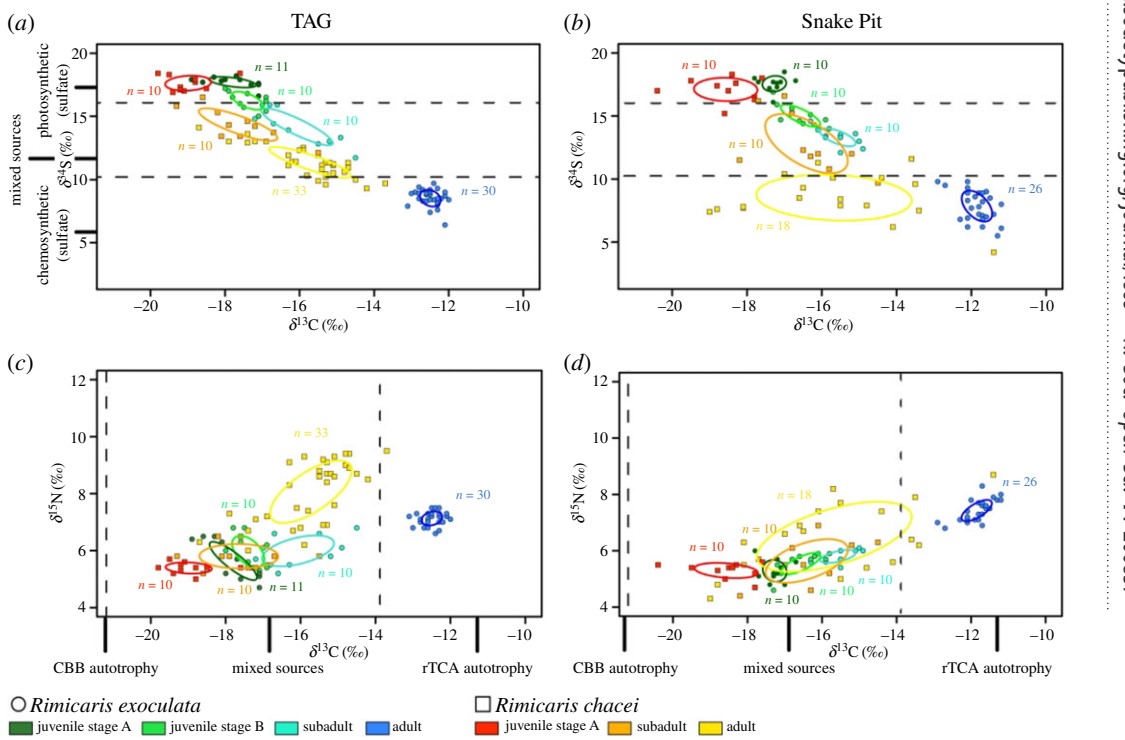

**Figure 4.** Isotopic niches of *Rimicaris* spp. life stages. (*a*) TAG, carbon versus sulfur; (*b*) Snake Pit, carbon versus sulfur; (*c*) TAG, carbon versus nitrogen; (*d*) Snake pit, carbon versus nitrogen. Dotted lines denote the potential ranges of $\delta^{13}$C or $\delta^{34}$S values indicative of a source. Sulfur sources are indicated on the right and carbon sources below the panels.

with other vent fields in the MAR is the absence of *Bathymodiolus* mussel beds [30]. These mussels are recognized as ecosystem engineer species providing habitat for diverse macrofaunal and microbial communities enhancing thereby the local diversity of the vent [30–32]. As many *R. chacei* were collected within these mussel beds, their population could benefit from this larger range of available chemosynthetic resources. On the other hand, *R. chacei* inhabiting the TAG vent field do not have access to these chemosynthetic resources and probably feed on different resources to complete their diet, potentially some found outside or at the periphery of the chemosynthetic influence sphere of the field.

According to our results and to previous studies, $\delta^{15}$N values of *R. exoculata* adults are higher than those of other purely chemosymbiotic species from hydrothermal vents. In general, high $\delta^{15}$N values are associated with a high trophic position, as consumers present stepwise $^{15}$N-enrichment with each trophic level [12]. However, variations of $\delta^{15}$N in producers can also be related to different inorganic nitrogen sources such as nitrates ($\delta^{15}$N = 5–7‰) and ammonium ($\delta^{15}$N < 0‰) [35,36]. Metagenomic studies on the cephalothorax of *R. exoculata* revealed that *Campylobacterota*, unlike the other symbionts, possess nitrogen fixation pathways allowing them to use nitrate as a nitrogen source to produce ammonium [13]. Therefore, the relatively high $\delta^{15}$N values of *R. exoculata* adults could also be attributed to a higher supply from the *Campylobacterota* than from other symbionts. These results

Isotopic niches of *R. exoculata* adults were also distinct between the two vent fields, with a slight $^{13}$C-enrichment at Snake Pit. This could be due to local variations between the two vent fields in isotopic fractionation of the dissolved inorganic carbon (DIC) used by chemoautotrophic primary producers, as observed in other hydrothermal vent systems [10]. These variations could also be the result of a slightly more important contribution from CBB-fixing symbionts, either the *Gammaproteobacteria* or the iron oxidizing *Zetaproteobacteria* [13], to diets of *R. exoculata* from TAG. A more important contribution of *Zetaproteobacteria* at this vent field would also fit with TAG fluid chemistry, richer in iron [33], and with the higher abundance of this group in bacterial assemblages associated with *R. exoculata* eggs at this site compared with Snake Pit [34]. These results nonetheless contrast with previous studies that did not report variations in the isotopic compositions *R. exoculata* from different vent fields [17].

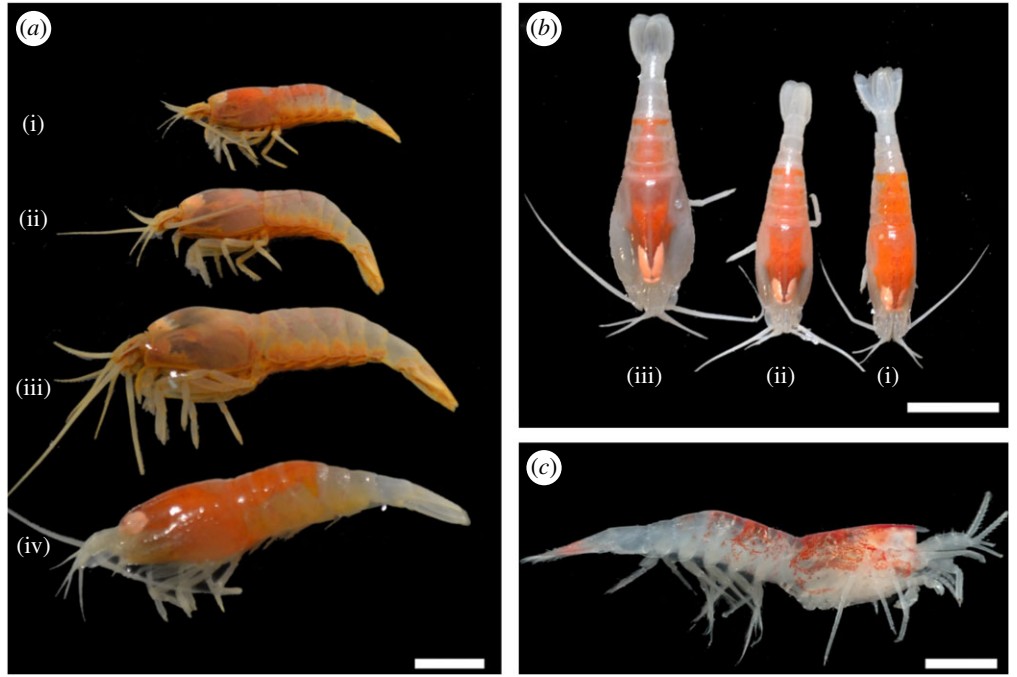

**Figure 5.** Juvenile stages of alvinocaridids from Mid-Atlantic Ridge vents according to the revised classification. (*a*) Different life stages of *R. chacei*: (i) stage A juvenile, (ii) stage B juvenile (or subadult) and (iii) small adult, compared with (iv) an early stage A juvenile of *R. exoculata*. Scale bar = 5 mm. (*b*) Different juvenile stages of *R. exoculata*: (i) stage A juvenile, (ii) stage B juvenile, (iv) stage C (or subadult). Scale bar = 10 mm. (*c*) A small individual of *Mirocaris fortunata*, morphologically similar to stage A juveniles of *R. chacei* and found in the same locations. Scale bar = 2 mm.

highlight that, in hydrothermal vent ecosystems, variations in inorganic nitrogen use at the base of food webs should be taken into account when trying to estimate animals' trophic positions. Indeed, even if many free-living *Campylobacterota* do not have this ability to use nitrate as a nitrogen source [37,38], others, such as the mat-forming *Sulfurovum* sp., do [39]. Moreover, sequences close to those of *R. exoculata Campylobacterota* symbionts have been retrieved from the environment in several MAR vent sites [14,40], where they could be an important food source for the vent fauna.

## 4.3. What can we learn about the *Rimicaris* life history?

Since the isotopic composition of animals is partially driven by their habitat use [41,42], analysing stable isotope ratios of recent immigrants can provide a record of their migratory history if the environments they have encountered have sufficiently distinct isotopic landscapes [43]. In hydrothermal vents, the large difference in $\delta^{34}S$ existing between seawater sulfate and vent fluid sulfides has been used to discriminate organic matter of photosynthetic origin (approx. 16‰ to 19‰) from that produced by local chemosynthesis (sulfide oxidation; −9‰ to 10‰) [9,10].

In our study, most alvinocaridid adults had $\delta^{34}S$ values under the 10‰ threshold, indicating a strong reliance on sulfur oxidizing microbial production. Conversely, almost every earlier stage of *Rimicaris* juvenile showed $\delta^{34}S$ values higher than 16‰, consistent with a nutrition directly or indirectly based on epipelagic photosynthetic production. Previous work on storage lipids of *R. exoculata* juveniles from the MAR already suggested that photosynthesis-derived carbon is an important part of their diet [19]. These studies were based on typical photosynthetic lipid markers—like C20:5n3 or C22:6n3 fatty acids—and $\delta^{13}C$ values. However, recent findings have shown that some of these lipid markers could also be biosynthesized by deep-sea bacteria [44]. This bacterial origin was also proposed for the carotenoid pigments that confer the red/orange storage lipid colour in *R. exoculata* [45].

Here, sulfur isotope ratios are consistent with a trophic shift between the ontogenic stages of *R. exoculata* and provide additional support to the hypothesis of bathypelagic-feeding life stages outside the vent field before juvenile recruitment. Moreover, ontogenic $\delta^{34}S$ variations observed in *R. chacei* indicate a similar trophic shift between photosynthesis-based nutrition in juveniles to mostly thiotrophy in adults. The gradual increase in $\delta^{13}C$ and, to a lesser extent, in $\delta^{15}N$ from juveniles to

adults further strengthens the hypothesis of an ontogenetic nutritional shift coinciding with the anatomical changes observed between these life stages [17]. Overall, the isotopic compositions of the intermediate stages (stage B juveniles and subadults) are compatible with a mixed diet containing both photosynthetic and chemosynthetic food sources. These could also reflect the gradual replacement of tissues built from photosynthesis-derived production during larval life outside the vent field, by tissue newly synthesized from organic matter of symbiotic chemosynthetic origin during juvenile growth and development. Still, without information on the spatial foraging behaviour of these animals, nor symbiont activity, the hypothesis of a mixed diet, maintaining sunken photosynthetic uptake in addition to chemosynthesis, remains valid because shrimp at these life stages might move to and from the vent fields and feed in contrasted habitats.

Trophic shifts following recruitment and habitat change have been reported in other vent shrimps, including *R. kairei* from vents on the Central Indian Ridge [46], *Opaepele loihi* and *Alvinocaris marimonte* from vents of the Mariana Arc [47], as well as in the mussel *Bathymodiolus azoricus* from the MAR [48]. In all these species, new immigrants typically had $\delta^{13}$C around −17‰ to −19‰, which reflects feeding on photosynthetically derived materials. In mussels, trophic shift occurred soon after recruitment, at very small sizes while growth had barely started. In bathymodiolin species, larvae do not host symbionts, but these are acquired from the environment soon after recruitment [49], thus coinciding with the early isotopic shifts. Symbiont-related lineages have been found on microbial communities growing on eggs during maternal brooding [34], but it is still not known whether part of these bacteria are retained by larvae after hatching. Our isotopic results with typical photosynthetic signature advocate that if some symbiotic bacteria are transported by larvae during their planktonic phase, these are probably not actively participating in larval metabolism or use specific metabolic pathways that are currently not characterized. Symbiont development and timing of acquisition (i.e. before or after dispersal) remain open questions whose investigation will probably allow a better understanding of the origin of the ontogenetic shift.

No conclusion can currently be drawn about where exactly in the water column the larvae may live. They may stay close to the seafloor with other deep-sea epibenthic fauna, or live several hundreds of metres above the vent fields, while consuming, in both scenarios, organic carbon sinking down from the surface. Recently, Nomaki *et al*. [50] revealed that natural abundance radiocarbon content ($\Delta^{14}$C) clearly shifted from lower ratios in organisms living in bottom waters or at vents to higher ratios in water column plankton in the area of the Izu-Ogasawara Arc. $\Delta^{14}$C could therefore be used as an ecological tracer to provide additional hints on the habitat of dispersing *Rimicaris* larvae. In any case, they most likely reach the pelagic environment before the end of their dispersal as *Rimicaris* 'post-larvae'—no precise species identification—have been collected between 1990 and 3060 m depth above the MAR [51]. Because of their higher productivity and larger food availability, combined with the fact that crustacean larvae are in general strong swimmers in comparison with other faunal groups [52], it is tempting to hypothesize a larval migration up to the photic zone. However, it is unclear if these larvae could tolerate the lower pressure and higher temperature prevailing near ocean surface. For instance, *M. fortunata* hatched larvae, collected from a shallower vent field, survived in a good physiological state at atmospheric pressure and 10°C, but not at 20°C in the same pressure conditions [53]. Therefore, alvinocaridid migration towards the surface during larval dispersal might be limited by their physiological tolerance to pressure and temperature conditions.

While we have no direct evidence concerning the exact location of *Rimicaris* larval stages during their planktonic life, we observed a clear separation between the isotopic niches of *R. exoculata* and *R. chacei* stage A juveniles, with more $^{13}$C-depleted values for *R. chacei* (figure 4). This separation could arise from $\delta^{13}$C variations linked to bionomic (resource-related) or/and scenopoetic (habitat-related) factors during the planktonic life of each species [41,42]. In either case, it suggests that *R. exoculata* and *R. chacei* are likely to present different larval life histories, with different spatial distributions and/or different feeding habits. In addition, our morphological reassessment revealed a large difference in the size at recruitment (table 1), which also strongly advocates for distinct larval dispersal histories between the two species.

# 5. Conclusion

Our present study provides a good illustration of the identification issues that persist in deep-sea ecosystems, even in the best-known species of these environments, like *R. exoculata*. As exemplified by this study, such misidentifications are not solely small details of interest for expert taxonomists. The interspecific differences we highlighted would otherwise have been, through confusion between the juveniles of these two congeneric

species, interpreted as ontogenic variations of *R. exoculata*. This could then be an important source of bias when attempting to understand fundamental aspects of the ecology and the biology of these species, including but not limited to their nutrition and life-history traits. These issues also preclude an accurate overview of the biodiversity of these ecosystems, and of our planet in general. We reiterate here the recent call by Glover *et al.* [54] for an integrative taxonomy, maintaining traditional taxonomic procedures but coupling them to other insights about species' biology, and keeping in mind that previous identifications should be considered as current hypotheses, open to reassessment when necessary, like any other scientific result.

Ethics. The animals used in this study were collected on the French contract area for the exploration of polymetallic sulfur deposits and are not considered as endangered species. Each individual was immediately preserved in ethanol or frozen at −80°C after recovery and no live experiments were carried out.

Data accessibility. All sequences have been deposited in GenBank under accession nos. MT270699–MT270782 for each juvenile individual. Dataset compiling isotopic ratios of each shrimp individual is available in the SEANOE database at: https://doi.org/10.17882/74512.

Authors' contributions. P.M. collected the specimens in 2018, carried out the taxonomic and molecular laboratory work, prepared samples for stable isotope analysis, led data analysis, participated in the conception and design of the study and drafted the manuscript; L.N.M. carried out the isotopic analyses, provided his help for the statistical analyses of isotopic data and critically revised the manuscript; M.S. provided his help for the taxonomic analysis and critically revised the manuscript; M.-A.C.-B. conceived and designed the study, coordinated the study and helped draft the manuscript; F.P. collected the specimens in 2017, helped with the analysis of the sequences, conceived and designed the study, coordinated the study and helped draft the manuscript. All authors gave final approval for publication and agreed to be held accountable for the work performed therein.

Competing interests. We declare we have no competing interests.

Funding. This work was supported by the Ifremer REMIMA programme and the Region Bretagne ARED funding.

Acknowledgements. The authors thank the captains and crews of the R/V *Pourquoi pas*? and the Nautile submersible team for their efficiency, as well as the chief scientists and scientific parties of the HERMINE (http://dx.doi.org/10.17600/17000200) and BICOSE 2 cruises (http://dx.doi.org/10.17600/18000004). Additional thanks go to Dr. Gilles Lepoint (SIESTE workgroup, Laboratory of Oceanology, University of Liège) for facilitating our access to ULiège's stable isotope facility. We also thank Ouafae Rouxel for her help with the molecular barcoding work at the laboratory. We are also grateful to Dr. Laure Corbari and Lowick MNR for the pictures of the alvinocaridid juveniles that they took on the BICOSE 2 expedition and which were used in the second figure of this article.

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
