## [Reviewer comments · Royal Society Open Science]

Review History

RSOS-200837.R0 (Original submission)

Review form: Reviewer 1

Is the manuscript scientifically sound in its present form?

Yes

Are the interpretations and conclusions justified by the results?

Yes

Is the language acceptable?

Yes

Do you have any ethical concerns with this paper?

No

Have you any concerns about statistical analyses in this paper?

No

Recommendation?

Accepted with minor revision (please list in comments)

Comments to the Author(s)

I have previously reviewed the manuscript by Methou et al. entitled "Integrative taxonomy revisits the ontogeny and trophic niches of *Rimicaris* vent shrimps" for Proceedings B, and I am glad to see a revised version in Royal Society Open Science. The manuscript is an integrative study which presents new and important data for two *Rimicaris* shrimps living in hydrothermal vents. Particularly of interest is the isotopic data revealing trophic shifts and insights on their life history. This revised version has adequately addressed the concern from my previous review, with really great efforts by the authors to improve both text and display items. This paper will be essential and highly useful for future workers who wishes to undertake research in hydrothermal vent shrimps. I therefore recommend publication of this manuscript in Royal Society Open Science, following considerations of final few minor points for readability.

Minor points:

Throughout: I suggest not starting a sentence with abbreviated genus (ie. R.) but write out the genus in full (ie. *Rimicaris*).

Line 119: Seems to be an unnecessary space in the 1R primer.

Line 126 and elsewhere: It seems like each of your GenBank accession only contains a single sequence. In that case, you can delete the '.1' at the end.

Line 178: Using dashes make it a little bit confusing as to what subadult refers to. Maybe just use parentheses?

Line 248: No need to capitalize Alvinocaridid – only if Alvinocarididae.

Line 311: Do not italicize 'sp.'

Line 346: Capitalize Central and Ridge

Line 383: Extra dot after table 1

Table 1: et should be &; lipid storage should have capital L; Some can be deleted leaving just Morphological characteristics

Figure 1: Legend, et should be &

Figure 4: The light green text is difficult to read and may be improved.

Figure 5: The blank spaces in parts A and C above and below the shrimps mean the figure can be made smaller vertically by better trimming.

Decision letter (RSOS-200837.R0)

Dear Dr Methou

On behalf of the Editors, I am pleased to inform you that your Manuscript RSOS-200837 entitled "Integrative taxonomy revisits the ontogeny and trophic niches of *Rimicaris* vent shrimps" has been accepted for publication in Royal Society Open Science subject to minor revision in accordance with the referee suggestions. Please find the referees' comments at the end of this email.

The reviewers and handling editors have recommended publication, but also suggest some minor revisions to your manuscript. Therefore, I invite you to respond to the comments and revise your manuscript.

- Ethics statement

- Data accessibility

<http://datadryad.org/submit?journalID=RSOS&manu=RSOS-200837>

- Competing interests

- Authors' contributions

- Acknowledgements

- Funding statement

Because the schedule for publication is very tight, it is a condition of publication that you submit the revised version of your manuscript before 19-Jun-2020. Please note that the revision deadline will expire at 00.00am on this date. If you do not think you will be able to meet this date please let me know immediately.

If your manuscript is newly submitted and subsequently accepted for publication, you will be asked to pay the article processing charge, unless you request a waiver and this is approved by Royal Society Publishing. You can find out more about the charges at

<https://royalsocietypublishing.org/rsos/charges>. Should you have any queries, please contact openscience@royalsociety.org.

Kind regards,

Anita Kristiansen
Editorial Coordinator

on behalf of Dr Punidan Jeyasingh (Associate Editor) and Kevin Padian (Subject Editor)
openscience@royalsociety.org

Associate Editor Comments to Author (Dr Punidan Jeyasingh):

Comments to the Author:

This ms was reviewed by two experts at ProcB earlier. I thank the authors for submitting this ms to RSOS. The authors have addressed all comments made by one of the reviewers at ProcB. Moreover, the authors responded to comments from another (more critical) reviewer. The revised manuscript was re-assessed by this expert. This reviewer is clearly satisfied with the revisions, although they have raised some minor (but useful/important) issues. With much gratitude to the expert reviewer, I am delighted to recommend this manuscript for publication after the authors have addressed these minor issues.

Reviewer comments to Author:

Reviewer: 1

Comments to the Author(s)

I have previously reviewed the manuscript by Methou et al. entitled "Integrative taxonomy revisits the ontogeny and trophic niches of *Rimicaris* vent shrimps" for Proceedings B, and I am glad to see a revised version in Royal Society Open Science. The manuscript is an integrative study which presents new and important data for two *Rimicaris* shrimps living in hydrothermal vents. Particularly of interest is the isotopic data revealing trophic shifts and insights on their life history. This revised version has adequately addressed the concern from my previous review, with really great efforts by the authors to improve both text and display items. This paper will be essential and highly useful for future workers who wishes to undertake research in hydrothermal vent shrimps. I therefore recommend publication of this manuscript in Royal Society Open Science, following considerations of final few minor points for readability.

Minor points:

Throughout: I suggest not starting a sentence with abbreviated genus (ie. R.) but write out the genus in full (ie. *Rimicaris*).

Line 119: Seems to be an unnecessary space in the 1R primer.

Line 126 and elsewhere: It seems like each of your GenBank accession only contains a single sequence. In that case, you can delete the '.1' at the end.

Line 178: Using dashes make it a little bit confusing as to what subadult refers to. Maybe just use parentheses?

Line 248: No need to capitalize *Alvinocaridid* – only if *Alvinocarididae*.

Line 311: Do not italicize 'sp.'

Line 346: Capitalize Central and Ridge

Line 383: Extra dot after table 1

Table 1: et should be &; lipid storage should have capital L; Some can be deleted leaving just Morphological characteristics

Figure 1: Legend, et should be &

Figure 4: The light green text is difficult to read and may be improved.

Figure 5: The blank spaces in parts A and C above and below the shrimps mean the figure can be made smaller vertically by better trimming.

Author's Response to Decision Letter for (RSOS-200837.R0)

See Appendix A.

Decision letter (RSOS-200837.R1)

Dear Dr Methou,

It is a pleasure to accept your manuscript entitled "Integrative taxonomy revisits the ontogeny and trophic niches of *Rimicaris* vent shrimps" in its current form for publication in Royal Society Open Science.

on behalf of Dr Punidan Jeyasingh (Associate Editor) and Kevin Padian (Subject Editor)
openscience@royalsociety.org

Follow Royal Society Publishing on Twitter: [@RSocPublishing](https://twitter.com/RSocPublishing)
Follow Royal Society Publishing on Facebook:
<https://www.facebook.com/RoyalSocietyPublishing.FanPage/>
Read Royal Society Publishing's blog: <https://blogs.royalsociety.org/publishing/>

Appendix A

Associate Editor Comments to Author (Dr Punidan Jeyasingh):

Comments to the Author:

This ms was reviewed by two experts at ProcB earlier. I thank the authors for submitting this ms to RSOS. The authors have addressed all comments made by one of the reviewers at ProcB. Moreover, the authors responded to comments from another (more critical) reviewer. The revised manuscript was re-assessed by this expert. This reviewer is clearly satisfied with the revisions, although they have raised some minor (but useful/important) issues. With much gratitude to the expert reviewer, I am delighted to recommend this manuscript for publication after the authors have addressed these minor issues.

Author's response: We are grateful to the editor for his acceptance of our manuscript for publication at the Royal Open Society Science journal. We have provided to our manuscript the additional minor corrections according to the recommendations of the first reviewer. We also provided the doi code of the isotopic dataset used in this study (SEANO database) according to the data accessibility conditions for publication at Royal Society Open Science.

Reviewer comments to Author:

Reviewer: 1

Comments to the Author(s)

I have previously reviewed the manuscript by Methou et al. entitled "Integrative taxonomy revisits the ontogeny and trophic niches of Rimicaris vent shrimps" for Proceedings B, and I am glad to see a revised version in Royal Society Open Science. The manuscript is an integrative study which presents new and important data for two Rimicaris shrimps living in hydrothermal vents. Particularly of interest is the isotopic data revealing trophic shifts and insights on their life history. This revised version has adequately addressed the concern from my previous review, with really great efforts by the authors to improve both text and display items. This paper will be essential and highly useful for future workers who wishes to undertake research in hydrothermal vent shrimps. I therefore recommend publication of this manuscript in Royal Society Open Science, following considerations of final few minor points for readability.

Minor points:

Throughout: I suggest not starting a sentence with abbreviated genus (ie. R.) but write out the genus in full (ie. Rimicaris).

Line 119: Seems to be an unnecessary space in the 1R primer.

Line 126 and elsewhere: It seems like each of your GenBank accession only contains a single sequence. In that case, you can delete the '.1' at the end.

Line 178: Using dashes make it a little bit confusing as to what subadult refers to. Maybe just use parentheses?

Line 248: No need to capitalize Alvinocaridid – only if Alvinocarididae.

Line 311: Do not italicize 'sp.'

Line 346: Capitalize Central and Ridge

Line 383: Extra dot after table 1

Author's response: All the minor corrections were made according to the recommendations of the referee.

Table 1: et should be &; lipid storage should have capital L; Some can be deleted leaving just Morphological characteristics

Figure 1: Legend, et should be &

Figure 4: The light green text is difficult to read and may be improved.

Figure 5: The blank spaces in parts A and C above and below the shrimps mean the figure can be made smaller vertically by better trimming.

Author's response: Minor corrections on figures and tables were made according to the recommendations of the referee.